# Dynamic Characteristics and Experimental Research of Linear-Arch Bi-Stable Piezoelectric Energy Harvester

**DOI:** 10.3390/mi13050814

**Published:** 2022-05-23

**Authors:** Xuhui Zhang, Fulin Zhu, Luyang Chen, Xiaoyu Chen, Yan Guo, Hengtao Xu

**Affiliations:** 1College of Mechanical Engineering, Xi’an University of Science and Technology, Xi’an 710054, China; 20205224060@stu.xust.edu.cn (F.Z.); chenluyang@stu.xust.edu.cn (L.C.); 19105016004@stu.xust.edu.cn (X.C.); 20205108039@stu.xust.edu.cn (Y.G.); 21205016039@stu.xust.edu.cn (H.X.); 2Shaanxi Key Laboratory of Mine Electromechanical Equipment Intelligent Monitoring, Xi’an University of Science and Technology, Xi’an 710054, China; 3College of Engineering, Zunyi Normal College, Zunyi 563006, China

**Keywords:** linear-arch, bi-stable, piezoelectric energy harvester, nonlinear

## Abstract

Collecting vibration energy in the environment is expected to solve the problem of the self-power supply of wireless monitoring nodes in underground coal mines. By introducing nonlinear factors, a linear-arch bi-stable piezoelectric energy harvester (LBPEH) is designed. In order to reveal the influence of system parameters on the dynamic characteristics of LBPEH, the magnetic force model is established by the magnetizing current method, and the restoring force model is acquired through experimental measurement. The electromechanical coupling dynamics model of the system is established based on the Lagrange equation and Kirchhoff’s law. The influence of excitation amplitude and excitation frequency on the dynamic characteristics of the piezoelectric energy harvester is simulated and analyzed. Moreover, experiments are designed to verify the results of the simulation. The results reveal that the restoring force of the linear-arch beam is nonlinear, and the LBPEH constructed by the linear-arch beam has an asymmetric potential well. Changing the excitation frequency or excitation amplitude can make the system in the well chaotic and achieve a large periodic motion state. With the increase of excitation amplitude, it is beneficial for the system to realize large periodic motion. The research provides theoretical guidance for the design of piezoelectric energy harvesters for different excitation conditions.

## 1. Introduction

In recent years, with the wide application of wireless sensor technology in all walks of life [1,2], its energy supply method has also attracted much attention. At present, most wireless sensor nodes are powered by chemical batteries. On the one hand, the service life of chemical batteries is limited, and frequent replacement of batteries will increase the cost; on the other hand, the massive use of chemical batteries will cause environmental pollution and other problems. Vibration energy in the environment is collected to power wireless sensor nodes, which is expected to solve the problem of self-power of wireless sensor nodes [3,4,5]. The piezoelectric vibration energy harvesting method is widely used because of its simple structure and high energy density.

The vibration energy in the environment is distributed in a wide frequency band, while the traditional linear piezoelectric energy harvester works in a narrow frequency band, so the energy collection efficiency is low and cannot meet the power consumption requirements of wireless sensor nodes [6]. Shahruz [7] arranged cantilever beam arrays with different resonant frequencies: research shows that under the action of external excitation, only one cantilever beam can match the vibration frequency of the environment, so the energy collection efficiency is low. In order to improve the energy harvesting efficiency of piezoelectric energy harvesters, researchers have carried out a lot of research on the introduction of nonlinear methods to broaden the working frequency band of energy harvesters [8,9,10,11]. Wang [12] proposed a cantilever beam with variable section thickness and established its distributed parameter model. Their research shows that the energy harvesting efficiency of the structure is increased by 78% compared with a cantilever beam with uniform thickness. Rezaei [13] increased the resonance bandwidth of the energy harvester by adding a spring at the end of the cantilever beam. Masana [14] designed a beam with variable axial stress, established its dynamic model and solved it analytically, the research results show that, with the increase of axial force, the output power and bandwidth of the collector both increase. Zhou [15] designed a new type of multimodal piezoelectric energy collector, and the research shows that in the frequency range of 3–300 Hz, the energy collection capacity of the structure was 25.5 times that of the structure without multimodal addition.

Among all kinds of nonlinear frequency extension methods, the introduction of magnetic force has been widely concerned because of its simple structure. The piezoelectric energy harvester can have multiple steady-state equilibrium positions by reasonably configuring the number of external magnets [16,17,18,19]. Zhou [20] designed a tri-stable piezoelectric energy harvester, the experimental results show that by adjusting the steering of the magnet, its working bandwidth can cover 4–22 Hz. Lan [21] coupled two cantilever beams with different resonant frequencies through magnetic coupling, the research results show that the energy output bandwidth and efficiency of the system are significantly improved. Ibrahim [22] constructed a bi-stable piezoelectric energy harvester by introducing nonlinear magnetic force and established its dynamic model. Their research results show that: under the external excitation condition of 0.5 g, compared with the linear piezoelectric energy harvester, the energy harvesting band of the structure is broadened by 35%. Shah [23] proposed a bi-stable piezoelectric energy harvester, established its finite element model and carried out experimental research, the result shows that compared with the linear piezoelectric energy harvester, the energy harvesting efficiency of this structure is increased by nearly 50%. The above piezoelectric energy harvester adopts a straight beam structure, which can only collect vibration energy in a single direction. However, the vibration energy in the environment is random and multi-directional, so the piezoelectric energy harvester with a straight beam structure cannot effectively collect energy in the environment. The curved beam structure has a large deformation after being excited by external excitation and can respond to the excitation in multiple directions, which has attracted a lot of scholars’ research [24,25]. Chen [26] proposed an arch-linear piezoelectric energy harvester and compared the static performance of the arch-linear beam and the straight beam through finite element simulation, the results show that under the same external force, the strain of the arch-linear beam is larger, and the output voltage is higher. Yang [27] proposed a piezoelectric energy harvester with an arc structure and established its theoretical analysis model; the experimental results show that the power generation of the arc structure beam is 2.55 times that of the straight beam.

Aiming at the characteristics of broadband, random and multi-directional vibration energy generated during the operation of underground coal mine equipment, in this paper, an LBPEH is proposed to efficiently collect vibration energy in coal mines. Using the magnetizing current method to establish its nonlinear magnetic model, the dynamic equation of piezoelectric energy harvester with asymmetric potential well characteristics is established considering the asymmetry of restoring force. The dynamic equation is solved using the Runge-Kutta algorithm, from the perspective of the time domain, the influence of excitation amplitude and excitation frequency on the dynamic characteristics of the system are analyzed. Finally, the correctness of theoretical analysis is proved by experiments. The research provides theoretical guidance and experimental verification for the design of bi-stable piezoelectric energy harvesters adapting to different external excitation parameters.

## 2. Structure and Mathematic Model of LBPEH

### 2.1. Structure of the LBPEH

Figure 1 shows the schematic diagram of the LBPEH. The system is mainly composed of a linear-arch beam, Polyvinylidence Fluoride (PVDF), a fixture and magnets. PVDF is evenly pasted on the surface of the linear-arch beam, and the external resistor R is connected to the electrodes at both ends of the piezoelectric material. Magnet A is fixed at the end of the linear-arch beam, magnet B is fixed on the fixture, and magnets A and B maintain a magnetic repulsion relationship. The horizontal distance between them is *d*, and the horizontal length of the linear-arch beam is *L*. When the linear-arch beam is subjected to external excitation along the y-direction, it will bend and deform; the PVDF pasted on the surface of the linear-arch beam will also be deformed. The output voltage amplitude of the system increases with the deformation of the linear-arch beam.

### 2.2. Magnetic Force Analysis

In order to establish the nonlinear magnetic model between magnets A and B, this paper uses the magnetizing current method to build the magnetic model. The geometric relationship between the magnets is shown in Figure 2.

The magnetization strengths of magnets A and B are $M_{a}$ and $M_{b}$, and their length, width, and height are $l_{a}$, $h_{a}$, $w_{a}$ and $l_{b}$, $h_{b}$, $w_{b}$, respectively. The center points of the upper and lower surfaces of magnet A are $O_{2}\;,O_{1}$. Due to the high stiffness of the linear-arch beam, the deflection angle is small under the action of external excitation, therefore, it can be approximated that ∠φ=∠AOB*,* $\mathrm{AO}=L+\frac{1}{2}l_{a}$. It can be known from the geometric relationship $\sin\varphi =\frac{\mathrm{AC}}{\mathrm{AO}}=\frac{y}{L+\frac{1}{2}l_{a}}$, $\cos\varphi =\sqrt{{1-\sin}^{2}\varphi }$. According to the magnetizing current theory, the magnitude of the magnetic force along the x-direction generated by magnet A by magnet B can be expressed as
(1)$$\begin{array}{cc} F_{m0}=\mu_{0}M_{a}S[H_{x1} & \left(y-\frac{h_{b}}{2}\cos\varphi ,d-\frac{h_{b}}{2}\cos\varphi ,0\right)\\ & -H_{x2}\left(y+\frac{h_{b}}{2}\cos\varphi ,d+\frac{h_{b}}{2}\cos\varphi ,0\right)] \end{array}$$
where $\mu_{0}\;$ represents the vacuum permeability; *S* represents the upper and lower surface area of the magnet A; $H_{x1}$ and $H_{x2}$ represent the magnetic field strength along the *y*-axis direction generated by magnet B on the lower surface and upper surface of magnet A, respectively, its expression is
(2)$$\begin{array}{cc} H_{x}=\frac{M_{b}}{4\pi }[tan^{-1} & \left(\frac{z_{p}y_{p}}{y\sqrt{z_{p}{}^{2}+y_{p}{}^{2}+x^{2}}}\right)+tan^{-1}\left(\frac{z_{n}y_{n}}{y\sqrt{z_{n}{}^{2}+y_{n}{}^{2}+x^{2}}}\right)\\ & -tan^{-1}\left(\frac{z_{p}y_{n}}{y\sqrt{z_{p}{}^{2}+y_{n}{}^{2}+x^{2}}}\right)-tan^{-1}\left(\frac{z_{n}y_{p}}{y\sqrt{z_{n}{}^{2}+y_{p}{}^{2}+x^{2}}}\right)] \end{array}$$
where $z_{p}=z+\frac{w_{b}}{2}$,$\;y_{p}=z+\frac{h_{b}}{2}$, $z_{n}=z-\frac{w_{b}}{2}$*,*
$y_{n}=z-\frac{hb}{2}$.

Since the magnetic force expression obtained by the magnetizing current method is complex, the magnetic force $F_{m}$ is simplified to a polynomial of the end displacement w(L,t) of the linear-arch beam by means of polynomial fitting
(3)$$F_{m}=c_{1}w\left(L,t\right)+c_{3}w{\left(L,t\right)}^{3}$$

### 2.3. Modeling of Restoring Force

Compared with traditional straight beams, the restoring force of the linear-arch beam is nonlinear due to the arch structure. It is complicated to establish the analytical expression of its restoring force through theoretical analysis, so this paper adopts the method of experimental measurement to obtain the restoring force model. During the experiment, a YLK-10 dynamometer was used to slowly push the end of the linear-arch beam, and the restoring force-displacement data were recorded. The average value of multiple measurements was taken, and MTALAB was used to fit the experimental data to obtain the expression of restoring force-displacement
(4)$$F_{r}=s_{3}w{\left(L,t\right)}^{3}+s_{2}w{\left(L,t\right)}^{2}+s_{1}w\left(L,t\right)$$

Figure 3 shows the experimental measurement and polynomial fitting results of the linear-arch beam, where $s_{3}$ = −56,681.227 $N/m^{3}$, $s_{2}$ = 127.293 $N/m^{2}$, $s_{1}$ = −13.99 N/m are constant coefficients on third, second and first-order terms, respectively. It can be seen from Figure 3 that the polynomial fitting curve is not symmetric about the origin, which is mainly because the response displacement of the arch structure is different when the curvature increases and the curvature decreases under the same external force.

### 2.4. Dynamic Model of LBPEH

In this paper, the Lagrange equation is used to establish the dynamic model of LBPEH
(5)$$L\left(x,t\right)=T_{S}+T_{P}+T_{M}+W_{P}+U_{m}-U_{r}$$

The following describes the specific expressions of various energies:

The kinetic energy of the metal base layer and the piezoelectric layer can be expressed as
(6)$$T_{S}=\frac{1}{2}\rho_{S}A_{S}\int_{0}^{L}{\left[\frac{\partial w\left(x,t\right)}{\partial x}+\dot{z}\left(t\right)\right]}^{2}dx$$
(7)$$T_{P}=\frac{1}{2}\rho_{P}A_{P}\int_{0}^{L}{\left[\frac{\partial w\left(x,t\right)}{\partial x}+\dot{z}\left(t\right)\right]}^{2}dx$$
where $\rho_{S}$, $\rho_{P}$ are the densities of the metal base layer and the piezoelectric layer, respectively; $A_{S}$, $A_{P}$ are the cross-sectional areas of the metal base layer and the piezoelectric layer, respectively; $\dot{z}\left(t\right)$ is the vibration velocity of the base.

The kinetic energy of the free end magnet can be expressed as
(8)$$T_{M}=\frac{1}{2}M_{t}{\left\{{\left[\frac{\partial \left(x,t\right)}{\partial t}\right]}_{x=L}+\dot{z}\left(t\right)\right\}}^{2}+\frac{1}{2}I_{t}{\left[\frac{\partial^{2}\omega \left(x,t\right)}{\partial t\partial x}\right]}_{x=L}^{2}$$
where $M_{t}$ is the mass of the magnet, and $I_{t}$ is the moment of inertia of the magnet.

The electrical energy generated by the piezoelectric layer can be expressed as
(9)$$W_{P}=\frac{1}{2}\int_{V_{P}}E_{3}D_{3}dV_{P}=\frac{1}{2}\int_{V_{P}}E_{3}\left(e_{31}S_{1}+\varepsilon_{33}^{S}E_{3}\right)dV_{P}$$
where $E_{3}$ and $D_{3}$ are the electric field strength and electric displacement, respectively; $e_{31}$ and $\varepsilon_{33}^{S}$ are the electromechanical coupling coefficient and dielectric constant, respectively.

According to the Rayleigh-Ritz principle, the vibration displacement ω(x,t) of any point on the beam in the y-direction can be expressed as
(10)$$w\left(x,t\right)=\sum_{i=1}^{n}\varphi_{i}\left(x\right)q_{i}\left(t\right)$$
where $\varphi_{i}\left(x\right)$ represents the *i*-th bending mode shape function of the beam, and $q_{i}\left(t\right)$ represents the *i*-th generalized modal coordinate. The linear-arch beam in this paper is mainly used for low-frequency environmental energy collection, so only its first-order vibration mode is considered. For a cantilever beam with one end fixed and one end free, its allowable function can be expressed as [28]
(11)$$\varphi \left(x\right)=1-\cos(\frac{\pi x}{2L})$$

Combined with the analysis of Equations (3) and (4), substituting Equations (6)–(9) into Equation (5) and combining with Equation (11), the electromechanical coupling dynamics equation of the system can be obtained
(12)$$\begin{array}{cc} M\ddot{q}\left(t\right)+C\dot{q}\left(t\right) & +\left(c_{3}-s_{3}\right)q^{3}\left(t\right)-s_{2}q^{2}\left(t\right)+\left(c_{1}-s_{1}\right)q\left(t\right)-\theta v\left(t\right)\\ & =-H_{S}\ddot{z}\left(t\right) \end{array}$$
(13)$$\theta \dot{q}\left(t\right)+C_{p}\dot{v}\left(t\right)+\frac{v\left(t\right)}{R}=0$$
where
(14)$$M=\left(\rho_{s}A_{s}+\rho_{p}A_{p}\right)\int_{0}^{L}\varphi^{2}\left(x\right)dx+M_{t}\varphi^{2}\left(L\right)$$
(15)$$H_{S}=\left(\rho_{s}A_{s}+\rho_{p}A_{p}\right)\int_{0}^{L}\varphi \left(x\right)dx+M_{t}\varphi \left(L\right)$$
(16)$$C_{p}=\frac{\varepsilon_{33}^{S}b_{p}L_{p}}{h_{p}}$$
(17)$$\ddot{z}\left(t\right)=Asin\left(wt\right)$$

## 3. Numerical Simulations

### 3.1. Static Bifurcation Characteristics and Potential Energy Analysis of the System

In order to analyze the influence of the system parameters of the bi-stable piezoelectric energy harvester on the response characteristics of the system, the static bifurcation and potential energy characteristics of the system are firstly analyzed, and the structural parameters of the linear-arch beam are shown in the Table 1.
(18)$$F_{m0}+F_{r}=0$$
(19)$$U=U_{m}-U_{r}$$

According to Equation (18), the static bifurcation of the system can be obtained, Figure 4a shows the static bifurcation results of the system. When the distance between the magnet at the free end and the magnet at the fixed end of the piezoelectric energy harvester is less than the distance between the bifurcation point $d_{0}$ = 23 mm, the system has two stable solutions and one unstable solution. With the increase of magnet distance, the distance between two stable solutions increases first and then decreases, and the system is a bi-stable system. When magnet distance *d* > $d_{0}$ = 23 mm, the system has only one stable solution, and the system is a monostable system. The stable solution is shifted upward, which is mainly due to the existence of the arch structure in the linear-arch beam leading to the deviation of its equilibrium position.

The potential energy of the system can be obtained from Equation (19). Figure 4b shows the potential energy diagram of the system at different magnet spacing, by changing the distance between two magnets, the magnetic force can be changed, and then the total potential energy of the system can be affected. When the distance between magnets is *d* = 13 mm, the system has a potential barrier and two deep potential wells, showing bi-stable characteristics. When the magnet spacing increases to *d* = 18 mm, the depth and width of the two potential wells become shallower and smaller. When the magnet spacing continues to increase to *d* = 23 mm, the potential energy curve of the system tends to be smooth, and the system transitions from bi-stable to monostable. When the magnet spacing is increased to *d* = 28 mm, the system does not have the characteristics of a bi-stable system and presents a single potential well, and the system becomes monostable. In addition, it can be observed from the potential energy diagram that when *d* = 13 mm and *d* = 18 mm, the potential energy curve of the system presents an asymmetrical phenomenon, and the potential well on the right is deeper than the potential well on the left, which is caused by the asymmetric restoring force of the system.

Combined with the static bifurcation characteristics of the system and the analysis results of potential energy, it can be seen that appropriate magnetic spacing can make the system have bi-stable characteristics and have a lower potential barrier, and the excitation frequency and excitation amplitude have an important impact on the dynamic characteristics of the system. Next, the influence of the two on the response characteristics of the system is analyzed.

### 3.2. Influence of Excitation Amplitude A on Dynamic Characteristics

Let *d* = 18 mm, *f* = 12 Hz, in order to explore the influence of excitation amplitude *A* on the dynamic characteristics of the system. Figure 5 shows a bifurcation diagram of the system with different excitation amplitudes. Figure 6 shows the phase diagram and time-displacement diagram of the piezoelectric energy harvester, when *A* is 5, 9, 12 and 18 $\;m/s^{2}$. It can be seen from Figure 5 that under different excitation amplitudes, the piezoelectric energy harvester will show different dynamic responses. When the excitation amplitude is within the range of 0–10$\;m/s^{2}$, the piezoelectric energy harvester moves in the well. When the excitation amplitude is in the range of 10–17$\;m/s^{2}$, the piezoelectric energy harvester is in chaotic motion. When the excitation amplitude is greater than 17$\;m/s^{2}$, the piezoelectric energy harvester will always be in a state of large periodic motion within the analyzed excitation amplitude range.

It can be seen from Figure 6a that when excitation amplitude *A* = 5$\;m/s^{2}$, the system is unable to cross the bi-stable barrier due to its small kinetic energy. At this time, the piezoelectric energy harvester can only vibrate in a single potential well, its vibration velocity is small, and the vibration displacement is only 6 mm. It can be seen from Figure 6b that when the excitation amplitude increases to *A* = 9$\;m/s^{2}$, the piezoelectric energy harvester crosses the bi-stable barrier and enters the deep potential well on the other side. However, since its kinetic energy is still small, the piezoelectric energy harvester is in a state of motion in the well with a period of two. By comparing Figure 6a and Figure 6b, it can be seen that the vibration displacement of the piezoelectric energy harvester increases from 6 mm to 10 mm as the system kinetic energy increases with increasing excitation amplitude. It can be seen from Figure 6c that when the excitation amplitude increases to *A* = 12$\;m/s^{2}$, the piezoelectric energy harvester is in a state of chaotic motion, and its response displacement amplitude is greatly improved compared with the state of motion in the well. The system has a high energy output; however, the piezoelectric energy harvester still exists the state of motion in the well, resulting in discontinuous high energy output of the system. If we continue to increase the excitation amplitude to *A* = 18$\;m/s^{2}$, and the kinetic energy of the system further increases, it can be seen from the Figure 6d that the piezoelectric energy harvester is in a state of large periodic motion at this time: the response speed and displacement reach the maximum, and the vibration displacement amplitude can reach 20 mm.

It can be seen from the above analysis that increasing the excitation amplitude will increase the kinetic energy obtained by the system, which will help the piezoelectric energy harvester to cross the bi-stable potential barrier and realize the movement between the wells.

### 3.3. Influence of Excitation Frequency f on Dynamic Characteristics

Let *d* = 18 mm, *A* = 12$\;m/s^{2}$, in order to explore the influence of the excitation frequency on the dynamic characteristics of the system. Figure 7 is the bifurcation diagram of the system at different excitation frequencies. It can be seen from Figure 7 that the frequency response characteristics of the system are very complex.

When the excitation frequency is in the range of 5.5 Hz, the piezoelectric energy harvester is in a state of chaotic motion. Figure 8a shows the phase diagram and time- displacement diagram when *f* = 2.5 Hz. In this case, the motion in and between the wells of the piezoelectric energy harvester exists simultaneously, and the system has high-energy output, but it is not continuous. When the excitation frequency is increased within the range of 5.5–10.5 Hz, the piezoelectric energy harvester is in a state of large periodic motion, compared with the chaotic motion state, the system has a continuous high-energy output at this time. It can be seen from Figure 8b that when the excitation frequency *f* = 8 Hz, the vibration displacement amplitude of the piezoelectric energy harvester can reach 15 mm.

If we continue to increase the excitation frequency, it can be seen from Figure 7 that the piezoelectric energy harvester will no longer make a large periodic motion. When the excitation frequency is in the range of 10.5–23.5 Hz, the piezoelectric energy harvester changes from a large periodic motion to a chaotic motion state. Figure 8c shows the phase diagram and time-displacement diagram of the piezoelectric energy harvester under the condition of *f* = 12 Hz. When the excitation frequency is greater than 23.5 Hz, the piezoelectric energy harvester moves in the well, and its response displacement amplitude and velocity decrease significantly. As can be seen from Figure 8d, when *f* = 25 Hz, the piezoelectric energy harvester can only vibrate in a single potential well, and the vibration displacement is only 1.5 mm. Based on the above analysis, it can be seen that by changing the frequency, the system will exhibit various nonlinear motion characteristics. When the excitation frequency is in the range of 23.5 Hz, the piezoelectric energy harvester can realize the motion between wells, and the system has high output. In particular, when the excitation frequency is in the range of 5.5–10.5 Hz, the piezoelectric energy harvester is in a large periodic motion, and the response displacement and velocity of the system are large.

Therefore, under the action of nonlinear magnetic force, the bi-stable piezoelectric energy harvester has a wide energy harvesting frequency band. Figure 8a,b shows that the velocity of the piezoelectric energy harvester is different at the two equilibrium points, and the phase diagram presents an asymmetrical phenomenon, which is caused by the asymmetrical restoring force of the linear-arch beam.

## 4. Experimental Validation

In order to verify the accuracy of the simulation analysis, an experimental platform was built for experimental verification. The experimental device is shown in Figure 9. It mainly includes the computer, vibration controller (VT-9008), power amplifier (GF-20), Exciter (E-JZK-5T), Laser Vibrometer (LV-S01) and vibration test analyzer (CoCo80-X). Figure 10 shows a linear-arch piezoelectric energy harvester consisting of a linear-arch beam, PVDF and a magnet. The PVDF is encapsulated with polyimide tape, and then evenly pasted on the upper surface of the linear-arch beam by epoxy resin. A magnet is glued to the ends of the linear-arch beam. In the experiment, a sinusoidal signal is set on the computer and sent out by the vibration controller. After being amplified by the power amplifier, it acts on the Exciter, so that the Exciter produces sinusoidal vibration, and the piezoelectric energy harvester is excited to vibrate. The terminal vibration velocity signal is measured by the laser vibrometer and stored in the vibration testing analyzer.

Figure 11 shows the phase diagram and time-displacement diagram of the piezoelectric energy harvester when *f* = 12 Hz, *A* = 5$\;m/s^{2}$ and *A* = 18 $m/s^{2}$. Figure 11a shows that when *A* = 5$\;m/s^{2}$, the kinetic energy of the system is small, and it cannot cross the bi-stable barrier. It is in the well motion state, and the vibration displacement is about 3 mm. When the excitation amplitude is increased to *A* = 18$\;m/s^{2}$, it can be seen from the Figure 12b that the piezoelectric energy harvester crosses the bi-stable potential barrier and exhibits bi-stable characteristics, and the piezoelectric energy harvester is in a state of large periodic motion. Compared with the motion in the well, the vibration velocity and displacement are significantly higher, and the vibration displacement amplitude is up to 17 mm.

Figure 12 shows the phase diagram and time-displacement diagram of the piezoelectric energy harvester, when *A* = 12$\;m/s^{2}$, *f* = 8 Hz and 12 Hz. Figure 12a shows that when *f* = 8 Hz, the piezoelectric energy harvester is in a large periodic motion state, and its vibration displacement is up to 20 mm. When the excitation frequency is increased to =12 Hz, it can be seen from Figure 12b that the piezoelectric energy harvester enters the chaotic motion state, and its vibration displacement decreases.

The experimental results are similar to the simulation results, but there are deviations mainly due to the fact that the simulation analysis of the piezoelectric energy harvester is the end of the vibration velocity. However, due to the existence of the arch part of the linear-arch beam in the vibration process, it will produce a deformation perpendicular to the velocity direction, and the position of the measured point of the laser vigrometer will be offset, leading to the deviation between the experimental results and the simulations.

## 5. Conclusions

In this paper, the electromechanical coupling dynamics model of the system is established for the LBPEH, and the model is solved by the Runge-Kutta algorithm. The influence of the excitation amplitude and excitation frequency in the system parameters on the dynamic characteristics is simulated and analyzed; the experiment verified the accuracy of the simulation results. The main conclusions are as follows:(1).By adjusting the magnet spacing *d*, a monostable or bi-stable system can be constructed. When *d* < *d*_0_ there are two stable solutions and one unstable solution, and the system behaves as bi-stable. When *d* > *d*_0_ the system has only one stable solution, and the system behaves as monostable. In addition, the stable solution is offset, which is due to the existence of an arch structure in the linear-arch beam, resulting in asymmetric restoring force of the linear-arch beam, which leads to the deviation of the stable equilibrium position.(2).With the increase of excitation amplitude, the kinetic energy obtained by the system increases, which is beneficial for the piezoelectric energy harvester to cross the bi-stable barrier and realize inter-well motion, and the response displacement and velocity of the piezoelectric energy harvester will increase.(3).Changing the excitation frequency can make the system exist in different motion states. When the excitation amplitude *A* = 12 m/s^2^, the piezoelectric energy harvester will have a large periodic motion in the low frequency range, and the frequency band range is 5.5–10.5 Hz. Increasing the excitation frequency changes the system from a state of large periodic motion to a state of chaotic motion; when the excitation frequency is greater than 23.5 Hz, the piezoelectric energy harvester moves in the well.(4).When the piezoelectric energy harvester is in the chaotic motion state, the system has inter-well motion, and its response displacement amplitude and velocity increase significantly compared with the motion state in the well, which is conducive to realizing high energy output of the piezoelectric energy harvester. This provides a new idea for the design of piezoelectric energy harvester adapted to environmental excitation conditions.

## Figures and Tables

**Figure 1 micromachines-13-00814-f001:**
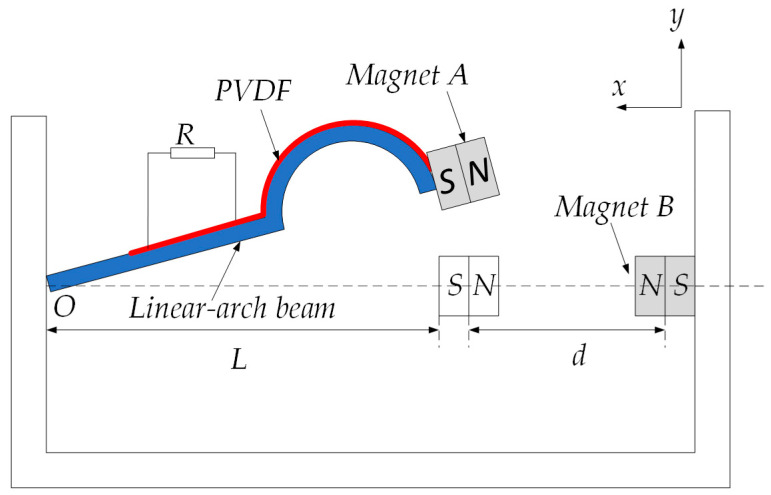
Schematic diagram of LBPEH.

**Figure 2 micromachines-13-00814-f002:**
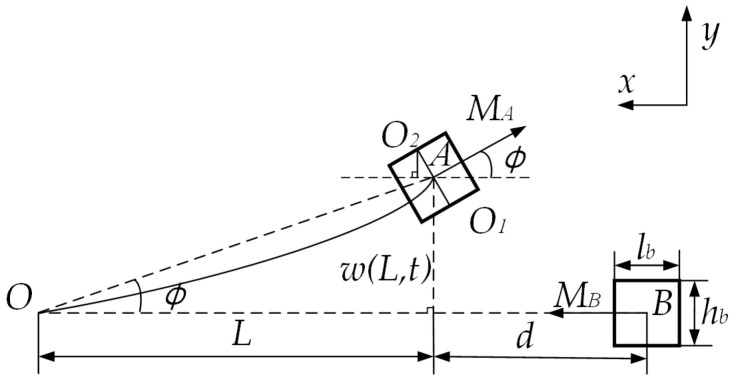
Magnetic force model.

**Figure 3 micromachines-13-00814-f003:**
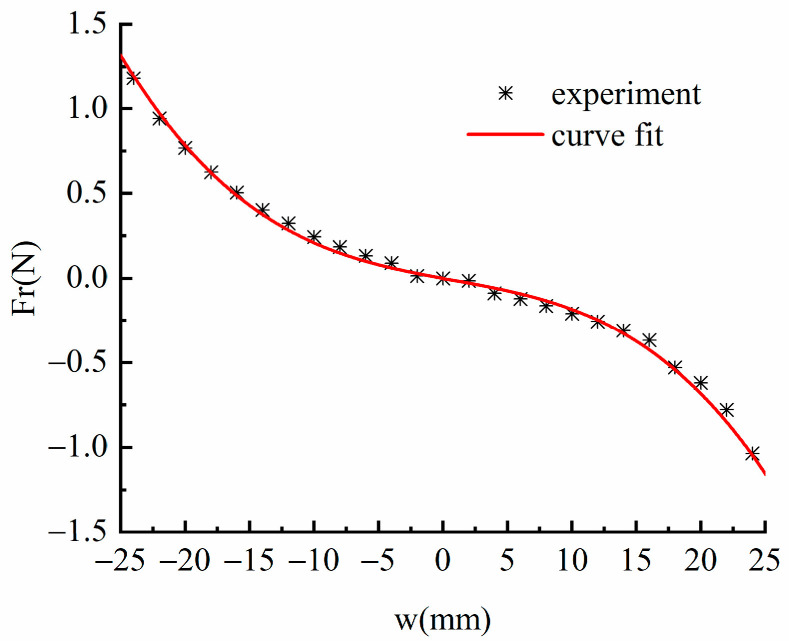
Restoring force model.

**Figure 4 micromachines-13-00814-f004:**
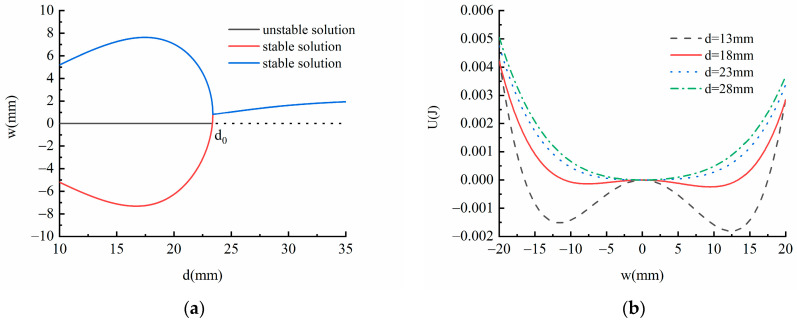
(**a**) static bifurcation of the system; (**b**) potential energy diagram of the system at different magnet spacing.

**Figure 5 micromachines-13-00814-f005:**
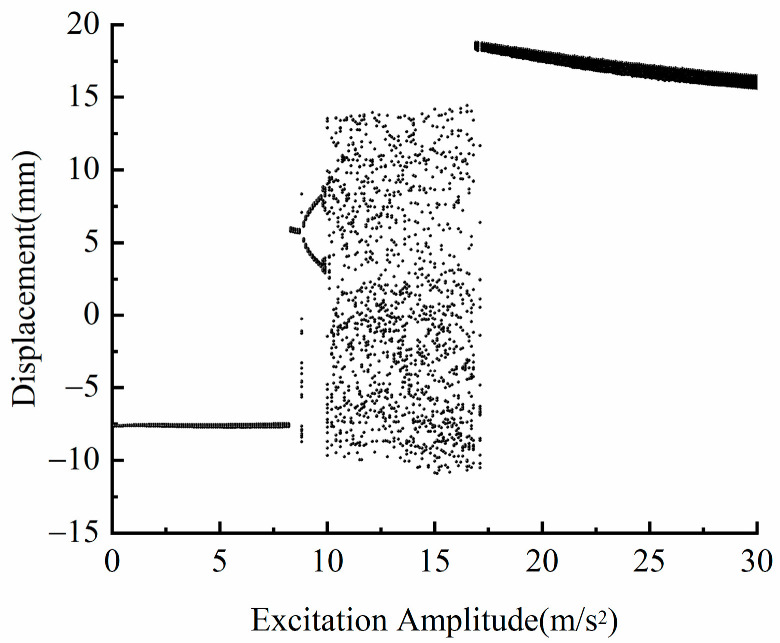
Bifurcation diagram of the system with different excitation amplitudes.

**Figure 6 micromachines-13-00814-f006:**
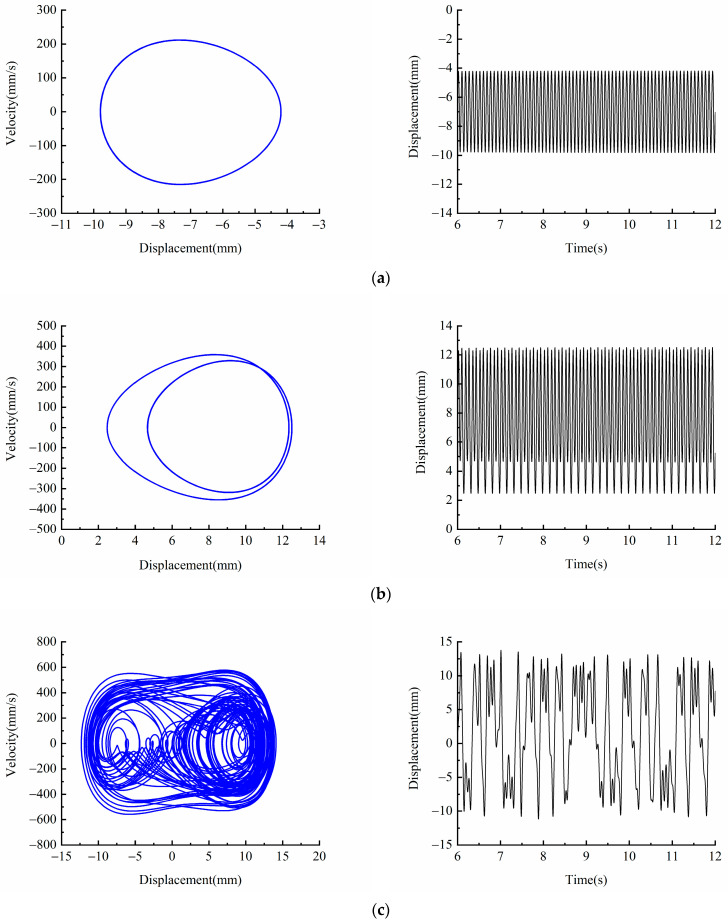
Phase portrait and time-displacement diagram of different excitation amplitudes *A*: (**a**) *A* = 5$\;m/s^{2}$; (**b**) *A* = 9$\;m/s^{2}$; (**c**) *A* = 12$\;m/s^{2}$; (**d**) *A* = 18$\;m/s^{2}$.

**Figure 7 micromachines-13-00814-f007:**
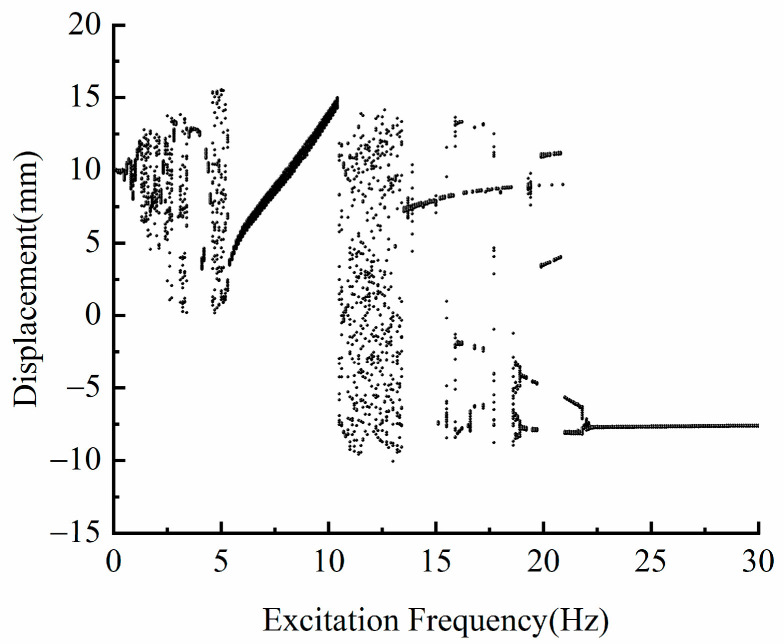
Bifurcation diagram of the system at different excitation frequencies.

**Figure 8 micromachines-13-00814-f008:**
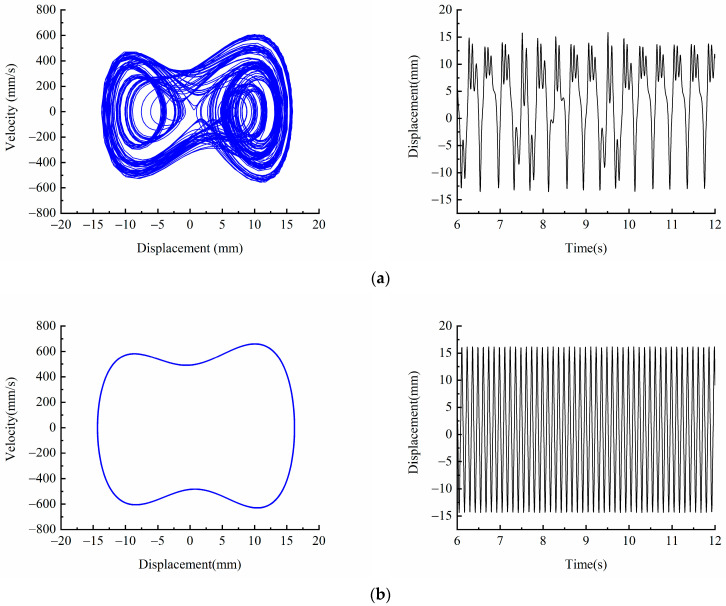
Phase portrait and time-displacement diagram of different excitation frequency *f*: (**a**) *f* = 2.5 Hz; (**b**) *f* = 8 Hz; (**c**) *f* = 12 Hz; (**d**) *f* = 25 Hz.

**Figure 9 micromachines-13-00814-f009:**
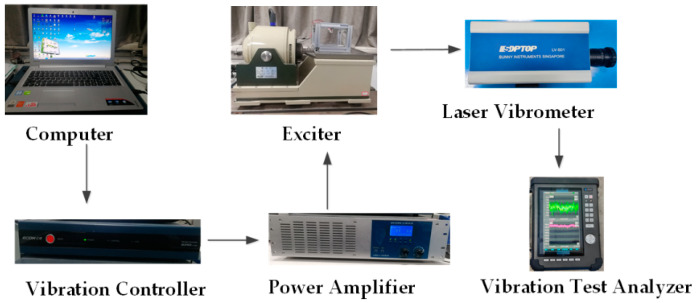
Experimental platform.

**Figure 10 micromachines-13-00814-f010:**
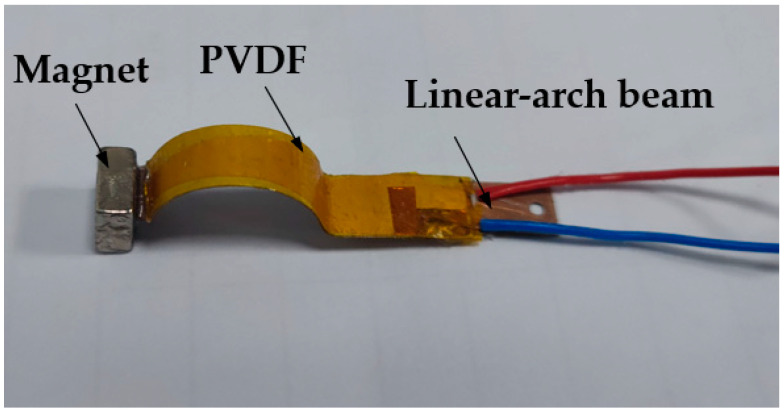
Linear-arch piezoelectric energy harvester.

**Figure 11 micromachines-13-00814-f011:**
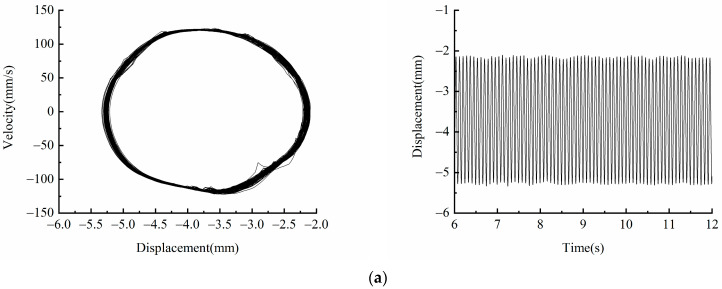
Experimental phase portrait and time-displacement diagram of different excitation amplitudes: (**a**) *A* = 5$\;m/s^{2}$; (**b**) *A* = 18$\;m/s^{2}$.

**Figure 12 micromachines-13-00814-f012:**
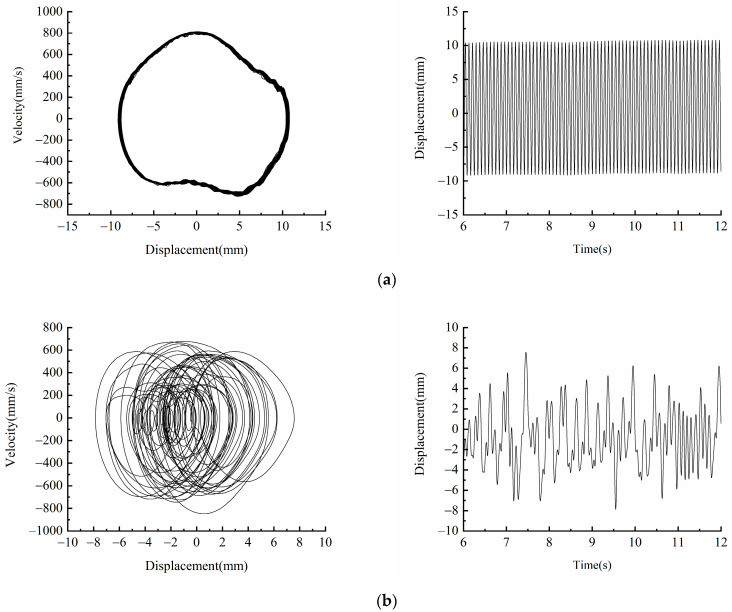
Experimental phase portrait and time-displacement diagram of different excitation frequency: (**a**) *f* = 8  Hz; (**b**) *f* = 12  Hz.

**Table 1 micromachines-13-00814-t001:** Structure parameters of LBPEH.

Parameter	Value	Parameter	Value
linear-arch beam		PVDF	
length/(mm)	40	length/(mm)	40
width/(mm)	8	width/(mm)	8
height/(mm)	0.2	height/(mm)	0.11
density/(kg/m^3^)	8300	density/(kg/m^3^)	1780
modulus/(GPa)	128	modulus/(GPa)	3
magnet		vacuum permeability/(H/m) magnetization/(A/m)	4π × 10^−7^ 4.5 × 10^5^
length/(mm)	5		
width/(mm)	10		
height/(mm)	10		
density/(kg/m^3^)	7500		

## Data Availability

Not applicable.

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
