# Peer review of "Dynamic Characteristics and Experimental Research of Linear-Arch Bi-Stable Piezoelectric Energy Harvester"

_micromachines, 2022, doi:10.3390/mi13050814_

Round 1

Reviewer 1 Report

The authors go through sufficient recent publications and provides clear evolution path of wide-band piezoelectric cantilever energy harvester. Arch structure offers higher energy efficiency and a better real world application compared to straight beam design. The design of experiments shows the displacement characteristics of frequency and speed separately. Overall, the draft is well-structured and the experimental results verified the numerical results. Below are some comments for the final adjustment.

  1. Figure 1: Why does the resistor only harvest at the linear part, instead of both arc and straight beam?
  2. Figure 2: symbol lb and hb format should align with paper content. 
  3. line 123: AOC or AOB?
  4. line 123 124: symbol la format should align with paper content. 
  5. line 195 to 209 is explaining d, not w. The authors could put the symbol d in the first few sentences, so the reader would not get confused. 

Reviewer 2 Report

  1. The problem that triggers the work need to be stated briefly and clearly in abstract.

  1. If there is any professional mathematic or engineering software used for mathematical modelling, please describe it in Section 2.

  1. It is hard to connect all the mentioned forces from section 2.2 to 2.4 with the performance of energy harvester. I believe that these forces are somehow closely related with it. However, it has not been described clearly.

  1. Eq (4) has not been defined in terms of its parameter.

  1. Experimental setup and flow should be discussed after section of Introduction.

  1. How to figure static bifurcation from Eq (17) and Figure 4?

  1. Figure 5 has not been analyzed thoroughly. In addition, Figure 5 is not clearly shown too. It is very vague.

  1. “Continue to increase the excitation amplitude to ?=18 ?/? 2 , 245 and the kinetic energy of the system further increases….”

Any equation can be used to explain this quoted statement?

  1. “The excitation amplitude will increase the kinetic energy obtained by the system, which will help the piezoelectric energy harvester to get rid of the magnetic constraint and realize the movement between the wells.”

Which result is used to justify this statement?

  1. Excitation amplitude has unit of m/s2. It is same unit as acceleration? Can we relate amplitude with acceleration?

  1. How to generate bifurcation diagram?

  1. Figure 11 and 12 have not been discussed thoroughly.

Reviewer 3 Report

The paper is well-written and prepared. It fits the scope of the journal and can be accepted after you address the following comments:

  • Usually, we perform the experiment and then build a model to understand it and verify the results. Please correct the sentence in line 18.
  • Could you please show detailed fabrication steps used to manufacture the device?
  • What is the bifurcation type in Figure 4 (a)? Is it saddle-node, pitch-fork...
  • The authors mentioned that changing the distance between the magnets changes the potential well. I believe the dual wells here are also due to the initial curvature, not the magnetic field alone. 
  • Change the x-axis label in Figure 5. (m/s2).
  • Figure 5 shows that as the excitation amplitude increases, the transition from lower well oscillation to the upper well oscillation leads to chaotic motion. Is there any way to avoid it during real operation because this is an undesirable zone to operate Energy harvesting?
  • Please show the transient response in the time-domain signals presented in Fig.6. This will help the reader to communicate with the manuscript easily and understand when the system settles down and goes to the steady-state response.
  • It was not clear if the authors planning to run the harvester at high excitation amplitudes or lower levels.
  • The experimental results are good.  if you show the chaotic response experimentally and the bifurcation diagram, it would be even nicer. 
  • Please revise the conclusion and re-organize it. 
